# Distribution and Organization of Descending Neurons in the Brain of Adult *Helicoverpa armigera* (Insecta)

**DOI:** 10.3390/insects14010063

**Published:** 2023-01-09

**Authors:** Xiaolan Liu, Shufang Yang, Longlong Sun, Guiying Xie, Wenbo Chen, Yang Liu, Guirong Wang, Xinming Yin, Xincheng Zhao

**Affiliations:** 1Henan International Joint Laboratory of Green Pest Control, College of Plant Protection, Henan Agricultural University, Zhengzhou 450002, China; 2State Key Laboratory for Biology of Plant Disease and Insect Pests, Institute of Plant Protection, Chinese Academy of Agricultural Sciences, Beijing 100193, China

**Keywords:** *Helicoverpa armigera*, backfilling, descending neuron, gnathal ganglion, posterior slope, three-dimensional reconstruction

## Abstract

**Simple Summary:**

One of the fundamental aims of neuroscience is to understand how the brain coordinates motor behaviors. Insects are perfect models for such studies because they have a simple but elaborate brain and a complex but stereotypical behavioral reservoir. The descending neurons (DNs) of insects connect the brain and thoracic ganglia and play essential roles in regulating insect behavior. A complete survey of DNs in the brain facilitates identifying the candidate neurons that may correlate with specific behaviors. In the present study, we comprehensively examined the distribution and organization of the DNs in the brain of a moth species, *Helicoverpa armigera*. The DN clusters are conserved across insect species. However, the cluster of DNd in *H. armigera* was not found in other studied species, and this cluster was only observed in males. This result suggests that the novel DNd cluster may consist of species- and sex-specific descending neurons. The innervation patterns of DNs in the brain are conserved across insect species, which indicates that the ventral part of the central brain plays multiple essential roles in triggering insect behaviors.

**Abstract:**

The descending neurons (DNs) of insects connect the brain and thoracic ganglia and play a key role in controlling insect behaviors. Here, a comprehensive investigation of the distribution and organization of the DNs in the brain of *Helicoverpa armigera* (Hübner) was made by using backfilling from the neck connective combined with immunostaining techniques. The maximum number of DN somata labeled in *H. armigera* was about 980 in males and 840 in females, indicating a sexual difference in DNs. All somata of DNs in *H. armigera* were classified into six different clusters, and the cluster of DNd was only found in males. The processes of stained neurons in *H. armigera* were mainly found in the ventral central brain, including in the posterior slope, ventral lateral protocerebrum, lateral accessory lobe, antennal mechanosensory and motor center, gnathal ganglion and other small periesophageal neuropils. These results indicate that the posterior ventral part of the brain is vital for regulating locomotion in insects. These findings provide a detailed description of DNs in the brain that could contribute to investigations on the neural mechanism of moth behaviors.

## 1. Introduction

Understanding how the brain coordinates motor behaviors is one of the fundamental aims of neuroscience. Ample research across insect species have demonstrated that a population of descending neurons (DNs), whose cell bodies are located in the brain and axons transmit the ‘‘commands’’ from the brain to the thoracic motor center, play a crucial role in controlling insect behavior [1,2,3]. For example, a descending neuron that functioned as a contralateral movement detector in the locusts could trigger the extension of the hind legs and lead to a jump [1]. In the cockroach *Periplaneta americana*, one descending neuron responding to antennal touch stimulation was shown to participate in escape behaviors [4,5]. In the cricket *Gryllus bimaculatus*, a calling sound-sensitive DN could adjust rotational or translational velocity during walking [6]. In the locust *Locusta migratoria*, several features detecting DNs were responsible for the specific deviations from routine flight [7,8]. In the flies, *Sarcophaga bullata* and *Calliphora erythrocephala*, some visually sensitive DNs were involved in flight movements by adjusting velocity, stabilization, and steering [2,9]. In the silkmoth *Bombyx mori*, some DNs have neurites that innervate the lateral accessory lobe (LAL) and exhibit activities in pheromone orientation [10]. In recent years, the powerful genetic tools for DNs in the fruit fly *Drosophila melanogaster* facilitated the functional examination of this type of neuron in various behaviors, including grooming, leg rubbing, and wing beating [11,12,13]. These studies have provided fundamental insights into the neural mechanisms underlying complex behavior control.

A systematic analysis of a large number of individual DNs of *D. melanogaster* was made recently using the available genetic tools, which provides a foundation for investigating how the brain coordinates motor behaviors across a wide range [14]. In addition to *D. melanogaster*, however, the comprehensive anatomical organization of DNs in the whole brain was only examined in *G. bimaculatus* and *P. americana*, by backfilling the axons from the neck connective [15,16]. The backfilling of the axons from the neck connective is an elegant approach for examining the DNs, which was evaluated in *D. melanogaster* compared with the labeling of genetic marks [17]. The roles of DNs may be directly related to the specific behavior of insects, and their morphologies and distributions are species-specific [18]. Therefore, exploring the anatomical distribution of DNs in more species is essential for understanding the neuroethological principles of insects.

The noctuid moth *Helicoverpa armigera* (Hübner) is a severe and polyphagous pest in the world [19,20]. Like other moths, odor-guided behavior is essential in locating host plants and mating partners for *H. armigera* [21,22,23]. The neural mechanism of olfactory behavior in *H. armigera* has been extensively studied from the peripheral to the central nervous system [23,24,25]. Previous studies in the *B. mori* had demonstrated that DNs showed responses to pheromones, which suggested that DNs mediate olfactory behaviors [3,26,27]. However, the detailed description of the number and distribution of DN somata in the moth brain remains unclear. In the present study, we investigated the number and distribution of DNs in the brain of *H. armigera*, by using backfilling via the neck connective combined with immunostaining techniques.

## 2. Materials and Methods

### 2.1. Experimental Animals

*H. armigera* were obtained from an established laboratory colony and kept in a climate incubator at 25 °C and 60% relative humidity under a long-day photoperiod regime (14/10 h light/dark). Adult *H. armigera* were fed daily with 10% sucrose solution after eclosion, and 2–4-day-old adults were used for the experiment.

### 2.2. Preparations and Backfill Labeling

The moth was restrained in a 1 mL plastic pipette tip with the head exposed and fixed with dental wax. The proboscis was stretched ventrally towards the abdomen and attached to the plastic pipette with wax. The head and neck scales were removed away by double-sided adhesive tape, and then the neck connective was exposed by opening the neck. Both neck connectives were cut and stained by filling them with Micro-ruby (4% tetramethylrhodamine dextran with biotin, Molecular Probes; Invitrogen, Eugene, OR, USA) crystal from the cut end. The preparation was placed in a moist environment at 4 °C overnight to allow the dye to diffuse. Then, the brain dissected out from the head in the Ringer’s solution containing 150 mM NaCl, 3 mM CaCl_2_, 3 mM KCl, 25 mM sucrose, and 10 mM N-tris(hydroxymethyl)-methyl-2-amino-ethanesulfonic acid (pH 6.9).

### 2.3. Immunocytochemistry

In order to visualize the structure of the brain, immunocytochemistry with an anti-synapsin antibody was performed. The dissected brains were fixed overnight at 4 °C with a 4% paraformaldehyde solution (PFA) in phosphate-buffered saline (PBS, pH 7.4) containing 684 mM NaCl, 13 mM KCl, 50.7 mM Na_2_HPO_4_, 5 mM KH_2_PO_4_. The brains were washed in PBS four times and each for 15 min at room temperature. The brains were preincubation in 5% normal goat serum (NGS, Sigma, St. Louis, MO, USA) in PBS containing 0.5% Triton X-100 (PBST; pH 7.4) for 3 h at room temperature to block the nonspecific staining. Then, the brains were incubated with the primary antibody anti-SYNORF1 (3C11, Developmental Studies Hybridoma Bank [DSHB], University of Iowa, Iowa City, IA, USA) in PBST, 1:100 for 5 days at 4 °C. Next, the brains were rinsed six times (20 min each) in PBS at room temperature and incubated with the secondary antibody Cy2-conjugated goat anti-mouse in PBST, 3:1000 for 3 days at 4 °C. After rinsing six times (each time 20 min) in PBS, the brains were dehydrated through an ascending ethanol series (50%, 70%, 90%, 95%, 99%, 2 × 100%, 10 min each). Finally, the brains were embedded in Permount after being cleared in methyl salicylate.

### 2.4. Confocal Microscopy and Image Acquisition

Serial optical images of the brain were obtained using confocal laser scanning microscopes (LSM980, Carl Zeiss, Jena, Germany; Nikon AIDH25, Japan) with × 10 and × 20 air objectives at a resolution of 1024 × 1024 pixels. The retrograde labeling with the Micro-ruby was excited by a HeNe laser 543-nm line, and the immunostaining for brain structures with Cy2 was excited by an Argon laser 488-nm line. The interval between the section was set at 3–5μm.

### 2.5. Data Analysis

The confocal image stacks were subjected to reconstruction using the Amira 5.3 software (Visage Imaging, Fürth, Germany) to visualize the three-dimensional anatomical organization of the stained descending neurons in the brain. Volume and diameter of the stained somata of descending neurons were measured by the “TissueStatistics” and “Measurement” tools, respectively. The number of somata of descending neurons was acquired by counting the number of cell body nuclei. Data about the number of each DNs cluster were calculated as the mean ± SD in the office 2019 software of Excel.

The images of neuronal processes of descending neurons in the brain from single optical sections and projection views were generated from confocal image stacks by using ImageJ 1.53c software (National Institutes of Health, Bethesda, MD, USA). The figures were prepared in Adobe Illustrator 2021 (Adobe Systems, San Jose, CA, USA). The color, contrast, and brightness of the image were adjusted in Image J.

### 2.6. Nomenclature

The general anatomy of *H. armigera* brain had been described previously by Chu et al. [28]. As Chu et al. proposed, the terminology and abbreviations for brain structures of *H. armigera* are based on the nomenclature suggested by Ito et al. [29]. The somata clusters were named based on their position by referring to the cluster names of *D. melanogaster* [14].

## 3. Results

The backfilling from the neck connective was performed on 25 preparations, and 19 were successfully stained (Figure 1A). The results revealed that a large number of somata of DNs and highly dense arborizations were stained (Figure 1B,C). The somata were mainly distributed around the central brain and grouped into different clusters (Figure 1(D1–D3)). The dense arborizations were mainly located in the ventral part of the central brain, including the gnathal ganglion (GNG), antennal mechanosensory and motor center (AMMC), posterior slope (PS), ventral protocerebrum, and the perioesophageal neuropils (Figure 1(E1–E3),(F1–F3)). The optic lobe was omitted in the illustration, where little evident labeling was found. A few arborizations were observed in the dorsal protocerebrum. However, the antennal lobe (AL), central body (CB), protocerebral bridge (PB), anterior optic tubercle (AOTU), and mushroom body (MB) showed no stained arborizations (Figure 1(E1–E3)).

### 3.1. Distribution and Number of Somata of DNs in H. armigera

#### 3.1.1. DNa Cluster

The DNa cluster is located in the anterior protocerebrum, medially to the AOTU and laterally to the lobes of MB (Figure 2A). All somata of this cluster are assembled. The mean number of stained DNa somata in each hemisphere of the protocerebrum is about 18 for males and 19 for females (Table 1). The mean diameter of somata in this cluster is about 16 µm for both males and females, and the mean volume of somata is 22.85 × 10^2^ µm^3^ for males and 22.04 × 10^2^ µm^3^ for females (Table 2).

The primary neurites of neurons of DNa form three bundles after originating from the somata (Figure 2B–F). Two thick bundles pass the area posterior to the MB lobes and lateral to the CB and converge into one bundle at the level of CB. And then, the bundle runs medially to the LAL and merges into the medial anterior PS. At the level of ventral CB, there is a prominent commissure linking both LALs observed (Figure 2C–E). The processes and pathways of DNa neurons to the ventral nerve cord (VNC) were unable to be further traced due to the highly dense labeling in the PS. The thin bundle runs anteriorly and ventrally through the region posterior to the AL and project to the periesophageal neuropil (Figure 2B–F). The thin bundle’s processes to the VNC could not be traced either.

#### 3.1.2. DNd Cluster

The DNd cluster is located in the dorsal superior lateral protocerebrum and at the entrance of the ocellar nerve entering the brain (Figure 3A–F). The mean number of stained DNd somata in each hemisphere of the protocerebrum is about 12 in males, whereas no DNd neurons were labeled in the females (Table 1; Figure 3G–I). The somata of DNd neurons are assembled into at least two sub-clusters, one located anterodorsally and one dorsally to the calyx (CA) (Figure 3B,C). The mean diameter of the somata in this cluster is about 11.57 µm, and the mean volume of somata is 12.16 × 10^2^ µm^3^ (Table 2).

The neurites of DNd neurons from the sub-cluster which is located anterodorsally to the CA converge into a bundle and project, bypassing the area anteromedial to the CA. Then, they exit the brain bypassing the area in the posteromedial PS. No collateral arborizations originating from these DNs in the brain were discerned. The neurites of DNd neurons from the other sub-cluster, which is located dorsally to the CA, run in a ventral direction bypassing an area anterior to the CA and bifurcating as two branches. One branch merges into the bundle formed by the DNd neurons, and the other projects ventrally to the posterior PS (Figure 3A–F).

#### 3.1.3. DNm Cluster

The DNm cluster is located in the middle of the superior medial protocerebrum and is clearly assembled into two clusters, DNm1 and DNm2 (Figure 4). DNm1 is located in the pars intercerebralis, and DNm2 in the posterior part of the pars intercerebralis. The two clusters are separated from each other. The mean number of stained DNm1 somata is about 28 for males and 26 for females, and in DNm2 is 31 and 33, respectively (Table 1). The mean diameter of DNm1 somata is about 11.91 µm for males and 11.27 for females, and the mean volume of somata is 13.35 × 10^2^ µm^3^ for males and 10.66 × 10^2^ µm^3^ for females (Table 2). In DNm2, the mean diameter of somata is about 13.10 µm for males and 12.65 µm for females, and the mean volume of somata is 17.07 × 10^2^ µm^3^ for males and 15.48 × 10^2^ µm^3^ for females (Table 2).

The neurites of DNm1 neurons form the median bundle and run in an anterior and ventral direction along the brain’s midline. The bundle splits into two tracts in front of the CB and projects further along the medial border of the periesophageal area to the VNC through the tritocerebrum (Figure 4A–F). The primary neurites of the DNm2 somata form several thin bundles and project ventrally to the posterior PS bypassing the area posterior to the CB (Figure 4G–L). Two notable bundles run through either side of the antler’s neuropil, cross the brain’s midline just above the esophagus foramen, and then project to the contralateral medial PS.

#### 3.1.4. DNp Cluster

The DNp cluster is located in the posterior protocerebrum, posteroventrally and posteromedially to the CA (Figure 5A–G). The mean number of stained DNp somata is about 196 for males and 189 for females (Table 1). The mean diameter of the somata in the DNp cluster is about 12.46 µm for males and 11.94 µm for females, and the mean volume of the somata is 15.63 × 10^2^ µm^3^ for males and 14.08 × 10^2^ µm^3^ for females (Table 2).

The neurites of many DNp neurons project to the posterior PS. The neurites from the cluster form at least three prominent bundles, and the bundles project to distinct neuropils (Figure 5C,D). Bundle 1 originated from the most lateral DNs in the DNp cluster in each hemisphere projects anteriorly and gives off collaterals to the anterior ventrolateral protocerebrum. Bundle 2 originated from the medial DNs of the DNp cluster in each hemisphere projects in an anterior direction. Then, the bundle turns laterally and runs down to the VNC bypassing the area medially to the anterior ventrolateral protocerebrum (AVLP). Bundle 3 originated from the dorsomedial DNs of the DNp cluster in each hemisphere projects in an anteromedial direction. Then, the bundle passes the CB’s dorsal surface and merges into the median bundle.

#### 3.1.5. DNv Cluster

The DNv cluster is located posteriorly to the AL. Usually, two or three DNv neurons form a sub-cluster, but the somata are clearly scattered in a larger area on the border between the protocerebrum and deutocerebrum (Figure 6A–E). The mean number of stained DNv somata in each hemisphere is about 15 for males and 14 for females (Table 1). The mean diameter of DNv somata is about 13.68 µm for males and 13.00 µm for females, and the mean volume of the somata is 18.01 × 10^2^ µm^3^ for males and 16.98 × 10^2^ µm^3^ for females (Table 2).

The neurites of DNv neurons run shortly into the PS and its neighboring neuropils, for instance, LAL, AMMC, and periesophageal neuropils. Their arborizations in distinct neuropils were impossible to be traced in the backfilled preparations.

#### 3.1.6. DNg Cluster

The DNg cluster is located in the medial and lateral cell body rind of the GNG (Figure 7A–H). The mean number of stained DNg somata is about 507 for males and 477 for females (Table 1). The mean diameter of DNg somata is about 14.13 µm for males and 13.14 µm for females, and the mean volume of somata is 19.86 × 10^2^ µm^3^ for males and 17.65 × 10^2^ µm^3^ for females (Table 2).

The neurites of DNg neurons and their projection directions were difficult to discern. In some preparations, the primary neurites and arborizations of a few stained DNg neurons were shown to be mainly confined in the GNG (Figure 7B,F).

### 3.2. Other Stained Neurons and Innervation Patterns in the Brain

In addition to the DNs, some long-fiber sensory neurons from peripheral sensory organs and the axons of ascending neurons (ANs) were stained simultaneously with DNs by backfilling from the neck connective in the moth brain (Figure 8 and Figure 9). These long-fiber sensory neurons originate from different sensory nerves, including the antennal nerve, frontal ganglion connective nerve, labral nerve, maxillary nerve, and labial nerve (Figure 8A–D). The somata for these long-fiber neurons were not observed in the brain but may be located in the peripheral sensory organs. For instance, a soma of the long-fiber neuron, here named DNx, was stained near the entrance of the antennal nerve entering the brain (Figure 8E). In the same preparation, neurons originating from the tegumentary nerve of the head were also found (Figure 8F,G). The somata of ANs were not observed in the brain, and the neuronal processes of ANs were difficult to distinguish from those of DNs since they were highly overlapping in the GNG and PS. However, two prominent ascending bundles in each hemisphere of the protocerebrum were observed projecting to the AVLP (Figure 9A–D). In addition, some stained arborizations were found in the neuropils, the accessory calyx (ACA) and the labial-palp pit organ glomerulus (LPOG) (Figure 9E,F). However, the origin of these arborizations from which type of neurons was impossible to determine. The processes in the LPOG may arise from the labial nerve (Figure 9G,H).

## 4. Discussion

The comprehensive description of the distribution and organization of descending neurons in the brain of adult *H. armigera* was made based on the staining by backfilling from the neck connective. The maximum number of DN somata in *H. armigera* is counted at about 980 for males and 840 for females, suggesting a sexual difference in DNs. There are 400 DNs for males and 340 for females in the cerebral brain, while 580 for males and 500 for females in the GNG. Therefore, a sexual difference may exist in the cerebral brain and GNG. Such sexual differences, however, have not been examined in other insect species. It has been demonstrated that there are 412 DNs in the cerebral brain of female *D. melanogaster*, 400 in female *G. bimaculatus* and 470 in male *P. americana* [15,16]. In GNG, there are 526 DNs in *D. melanogaster* and 300 in the locust *Schistocerca gregaria* and *L. migratoria* [17,30,31]. The numbers of DNs are different from one species to another.

### 4.1. DNa Cluster

We classified DN somata of *H. armigera* into six clusters, homologous in organization to those of other studied species. By comparing the location and projection of somata of DNs, the corresponding DNs clusters and numbers among different species are shown in Table 1.

The DNa is located in the anterior protocerebrum, medially to the AOTU and laterally to the lobe of MB. This cluster was also found in all studied insect species but named differently in different studies. For instance, it was called Group II in *B. mori*, AOTU cluster or DNa in *D. melaongaster*, LG in *S. gregaria*, i5 and i5n in *P. americana* and *G. bimaculatus* [3,15,16,17,26,27,32]. The number of DNa somata is about 21 in *H. armigera*, while it is 15 in the silkmoth, 38 in the fruit fly, 35 in the cockroach, and 22 in the cricket [10,15,16,17]. The mean diameter of somata in this cluster is about 16 µm, ranging from 10 to 25 µm in *H. armigera*. In the silkmoth, the somata have similar diameters ranging from 20 to 25 µm [10].

The DNa neurons give off arborizations in the ipsilateral neuropils, including the LAL, PS, ventral lateral protocerebrum (VLP), and GNG. The DNa neurons were classified into 6 types in the silkmoth and 10 types in the fruit fly based on the arborization patterns [10,14]. In general, DNa neurons have smooth processes in the PS and varicose processes in the GNG of the ipsilateral hemisphere. Group II neurons in *B. mori* showed responses to the pheromonal stimulation and exhibited flipflopping activity patterns between the left and right [10]. The descending neuron in the *G. bimaculatus* of B-DI1(1) belongs to the i5 cluster and shares the characteristic arborizations in the LAL, responding to the calling song [33]. The descending neurons of the flesh fly *S. bullata*, named DNDC3-6a/-6b, also seem to belong to the DNa neurons. However, DNDC3-6a/-6b have dendrites extending into the lobula of the optic lobe and AOTU. Such an innervation pattern was not observed in other studied species [34]. DNDC3-6a/-6b connects the male-specific visual system to the neck and flight motor for mediating the male-specific behavior, for instance, pursuing females. The electrophysiological responses of the DNa neurons were sex-specific in the studied species, e.g., *B. mori*, *G. bimaculatus*, and *S. bullata*. These results suggest that DNa neurons are involved in mediating sex-specific behaviors. Unlike those mentioned above, however, a DNa neuron found in *D. melanogaster*, named AX, was shown to have responses to looming and being involved in the rapid steering maneuvers of flying [35].

### 4.2. DNd Cluster

The DNd cluster is located in the dorsal part of the superior lateral protocerebrum, dorsoanteriorly to the CA, and closer to the entrance of the ocellar nerve into the brain. The DNd cluster was only observed in male *H. armigera*, not females. The maximum number of stained DNd neurons is about 15, which might account for the difference between the sexes in the number of DNs in the cerebral brain. No sexual difference was found in DNa, DNm, DNp, and DNv clusters. In addition, neurons homologous to the DNd of *H. armigera* were not found in the fruit fly, cockroach, and cricket [15,16,17]. Male *H. armigera* shows specific behavioral responses to female sex pheromones and possesses male-specific olfactory neuropils, the macroglomerular complex [23,36]. It would be interesting to investigate whether and how the DNd neurons are involved in male-specific olfaction. It is also interesting to investigate whether other moths possess such male-specific DNs.

### 4.3. DNm Cluster

The DNm cluster is located in the middle part of the superior medial protocerebrum and is clearly separated into DNm1 and DNm2 clusters. DNm1 is located in the pars intercerebralis, homologous to the cluster of PI in the fruit fly, cockroach, and cricket [15,16,17]. The number of DNm1 neurons is similar between the two sexes of *H. armigera* but is different across species. The maximum number in *D. melanogaster* is 48, while in the *P. americana* is 12 (Table 1). The neurites of DNm1 neurons of *H. armigera* join the median bundle and run in an anterior and ventral direction along the brain’s midline. The bundle bifurcates in front of the CB and then projects to the VNC through the tritocerebrum. A morphological study of the individual DNs in *D. melanogaster* revealed that there are two types in this cluster, DNc01 and DNc02 [14]. These neurons project through the median bundle and give off massive arborizations in large brain areas, including the optic lobes and the superior, lateral, and ventral protocerebrum. The widely spread patterns of the arborizations indicate that these neurons may play multiple roles in modulations.

DNm2 is located in the posterior part of the pars intercerebralis, clearly separated from DNm1. In *D. melanogaster*, DNm2 neurons were not reported, while *P. americana* and *G. bimaculatus* reported as i4 and c4 clusters [15,16,17]. The number and size of DNm2 neurons seem different across species. The maximum number of DNs in this cluster was about 35 in *H. armigera*, 52 in cockroach, and 34 in cricket (Table 1). The size of DNm2 neurons is 8–19 µm in diameter in *H. armigera*, and in cockroach is 10–30 µm [37].

The primary neurites of the DNm2 neurons project ventrally to the ipsilateral/contralateral PS. The descending neuron DN-IU1 of *L. migratoria*, homologous to the DNm2 neurons of *H. armigera*, extends numerous branches in the lateral protocerebrum and the dorsal deutocerebrum. The DN-IU1 neurons act as a deviation detector and show responses to various stimulations, for instance, light on/off stimuli, simulated course deviation (rotation of an artificial horizon), passive head rotation, frontal wind, and flight activity [38]. The descending neurons of D3, D4, and D5 of the cockroach are homologous neurons to the DNm2 and extend dense arborizations in the PS and tritocerebrum [37]. These neurons are higher-order ocellar interneurons and exhibit transient depolarizations to the ipsi- or contralateral ocellar illumination stimuli. In *S. gregaria*, polarization-sensitive descending neurons seem to be homologous to the DNm2 neurons; they have cell bodies in the posterior median pars intercerebralis at the level of the central body and protocerebral ramifications concentrated in the posterior protocerebrum near the posterior surface of the neuropil [39]. In *G. bimaculatus*, descending neurons of B-DI1(1), B-DI1(2), and B-DI1(3) seem to be homologous to DNm2 neurons as well, showing responses to antennal stimulation or ultrasound [33].

### 4.4. DNp Cluster

The DNp cluster is located in the posterior protocerebrum, posteroventral and posteromedial to the CA. This cluster is homologous to the clusters of SMP and DNp of *D. melanogaster*, and i1-i3, c1-c3 of *P. americana* and *G. bimaculatus* [14,15,16,17]. The maximum number of stained DNp somata is about 214 in *H. armigera*, while the fruit fly is about 325, the cockroach 320, and the cricket 300 (Table 1). The mean diameter of the somata in the DNp cluster is about 12 µm, and the biggest is about 18 µm. However, in the dragonfly *Aeschna grandis*, there are 3–4 large DN neurons in the same cluster; the diameters are about 60–80 µm. In the same cluster of *P. americana*, most giant DN neurons are about 35–50 µm [18]. The cervical giant fiber neuron (CGF) in *D. melanogaster* and the giant descending neuron (GDN) in flies *Musca* and *Calliphora* are also homologous neurons to the DNp of *H. armigera*. CGF is 20 µm in diameter, and the GDN is 30–50 µm [40,41]. These suggest that the size of the DNp somata varies in an extensive range across species.

The neurites of many DNp neurons in *H. armigera* project to the posterior PS. The morphological characterization of individual DNp neurons in *D. melanogaster* revealed that the arborizations of DNp neurons innervate diverse areas, including the PS, SMP, VLP, LAL, AMMC, tritocerebrum, lobula of the optic lobe (OL), and GNG [14]. A similar arborization pattern was also found in the homologous neurons, Group III, in *B. mori* [27]. The broad area of innervation indicates that the DNp neurons play multiple roles. The function of DNp neurons has been explored in several species, for instance, the locust, cockroach, dragonfly, and fruit fly. The descending movement detector, DCMD, is a homologous neuron to the DNp in the locust *Schistocerca vaga*, which has a large cell body (50 µm in diameter) and gives off some arborizations in the region posterior to the AL [42]. It responds to the movement of small contrasting objects in the visual field. In *P. americana*, the descending neurons of D1 and D2 are homologous neurons to the DNp, have large cell bodies (ca. 40 µm and 30 µm in diameter, respectively), and give off arborizations in the PS, tritocerebrum, and GNG. They show responses to the stimulation of illuminations to the ocelli [37]. The descending mechanosensory interneuron DMIa-1 in *P. americana* is also homologous to DNp neuron and extends arborizations posteriorly toward the dorsal surface of the deutocerebrum, dorsal up to, but not including, the antennal lobe [4]. It shows responses to mechanical stimulation of the contralateral antenna. In *G. bimaculatus*, a homologous neuron to DNp, DBNc2-1 has a large cell body (ca. 40 µm in diameter). It gives off arborizations in the lateral ocellar root and the non-glomerular neuropile of the dorsal lateral protocerebrum [6]. It shows responses to moving, grating, and lights. In the dragonflies *A. umbrosa* and *Anax junius*, eight DNp neurons have arborizations in the posterodorsal region of the brain and exhibit responses to movements of small target patterns [43]. In *S. gregaria*, ipsilaterally, polarization-sensitive descending neurons have somata near the posterior brain surface and arborizations concentrated in the posterior protocerebrum [39]. In *D. melanogaster*, the DNp neurons pIP10 for responding to the male calling song and P2b for responding to male courtship have somata in the posterior protocerebrum and arborizations in the lateral protocerebrum. They play roles by receiving input from the protocerebral neuron P1 [44,45]. The moonwalker descending neuron MDN of *D. melanogaster* also seems to belong to DNp neurons. It has a soma in the medial posterior protocerebrum and bilateral dendritic arborizations in the medial ventral protocerebrum and GNG. MDN controls backward walking when the fly encounters an impassable barrier [46]. A recent study found that olfactory stimuli induced backward locomotion was also regulated by this MDN neuron [47].

### 4.5. DNv Cluster

The DNv cluster is located on the border between the protocerebrum and deutocerebrum, and DNv somata are scattered in a large area posterior to the AL. The homologous cluster to DNv was also found in other species, for instance, PENP, DNb, and DNd in *D. melanogaster*, i6, i7, and c5-c7 in *P. americana* and *G. bimaculatus*, and Group I in *B. mori*. The maximum number of stained DNv somata in each hemisphere is about 17 in *H. armigera*, 80 in the fruit fly, 18 in the cockroach, and 27 in the cricket (Table 1). The diameter of DNv somata is 8–20 µm, which is similar to the measurements of *B. mori* [14,15,16,17,27].

The neurites of somata in DNv cluster project into the PS and neighboring neuropils, for instance, the LAL, AMMC, and periesophageal neuropils. Based on the arborization pattern, DN neurons in this cluster were classified into 9 types in *D. melanogaster*, and 4 in *B. mori* [14,27]. Many of the DNv neurons are contralateral. Their neurites project through the LAL commissure to the contralateral LAL or tritocerebral commissure to the contralateral hemisphere of the GNG. The tritocerebral commissure giant (TCG) in the *S. gregaria* is a DNv neuron responding to the wind on the head and light during flying [48,49]. Another similar DNv neuron has a smaller axon, called the tritocerebral commissure ‘dward’ (TCD) in *L. migratoria,* and shows responses to light, wind, and touch stimuli [50]. The descending neuron of B-DC1(5) in the *G. bimaculatus* belonging to this cluster shares the characteristic arborizations in the LAL and responds to calling songs [33]. Some tritocerebral descending neurons in flies *S. bullata* and *C. erythrocephala* are homologous neurons to the DNv. They have dendrites in the tritocerebrum, the AMMC, optic foci, and dorsal neuropils of the maxillary segment of the GNG and respond to wind stimulation and illumination [9]. In moths, including *B. mori* and sphinx moth *Manduca sexta*, the DNv neurons have smooth processes in the LAL and PS, and varicose processes in the PS and GNG of the contralateral hemisphere, and show responses to pheromones [10,51].

### 4.6. DNg Cluster

The DNg cluster is located in the medial and lateral cell body rind of the GNG. It is a large cell group, but it is difficult to classify them into sub-cluster since they are packed together. However, based on the arborization pattern of individual neurons, the DNg neurons were classified into 41 types [14]. Many DNg neurons give off arborizations in the ipsi- or contralateral neuropil of the GNG. Some DNg neurons also extend arborizations to other areas in the brain, including the tritocerebrum, AMMC, PS, and dorsal protocerebrum.

In *M. sexta*, a DNg neuron has a cell body in the ventral of the GNG, and the neurite crosses the midline of the GNG and gives off arborizations in a large field in the perioesphageal neuropil just below the foramen [51]. This neuron responded to both sex pheromone and light. In flies *S. bullata* or *C. erythrocephala*, a DNg has a cell body in the ventral GNG and gives rise to dendrites within the GNG, bilateral symmetrical on either side of the GNG midline. The electrophysiological recording showed this neuron responded to vibration to the head and simple stimuli of light on and off [9]. In *P. americana*, a descending mechanosensory interneuron, DMIb-1, has a cell body in the ventral cell body rind of the GNG. Its primary neurite crosses the midline and gives off many branches on either side of the midline. This neuron responds to direct mechanical stimulation of both antennae [4]. In *S. gregaria*, GNG descending neurons have cell bodies in the labial neuromere and arborizations restricted to a small dendritic tree and some beaded contralateral branches. These neurons show responses to polarization and moving grating [39]. A DNg neuron of *D. melanogaster*, named antennal descending neuron (aDN), is involved in control antennal grooming by receiving mechanosensory stimulation input from Johnston’s organ [11]. Two neurons, DNg11 and DNg12 of *D. melanogaster* also play essential roles in coordinating antennal grooming by initiating the front leg rubbing [12]. An unusual type of the GNG descending neurons, DNg02, a large population of neurons with 15 pairs in nearly homomorphic appearance, was involved in regulating wing motion by controlling both steering and thrust [13].

### 4.7. Innervation Patterns of DNs Cross Insect Species

As with other reported species, e.g., flies, locusts, cockroach, and cricket, the processes of stained neurons in the brain from the neck connective of *H. armigera* were mainly distributed in the ventral central brain, including the PS, VLP, LAL, AMMC, GNG, and other small periesophageal neuropils. A few arborizations were also found in the lobula of the OL, LPOG of the AL, and ACA of the MB. However, no arborizations were observed in some glomerular neuropils, e.g., the CA, pedunculus, and lobes of the MB, CB, PB, AOTU, and AL. These results indicate that the posterior ventral part of the brain is vital for regulating locomotion in insects. In contrast, many glomerular neuropils are not directly connected to the thoracic motor center. By analyzing the morphology of over 100 individual DNs obtained with genetic techniques in *Drosophila*, it was found that the PS contains the smooth processes of most DNs, and the processes are more in the inferior PS than in the superior PS [14]. These findings suggest that PS is the primary origin of DNs in insects. Furthermore, more than 90% of DN types of *D. melanogaster* have processes in the GNG, which might explain why the GNG is involved in various behaviors, including walking, stridulation, flight initiation, head movement, and respiration [14]. Morphological and physiological characterization of individual DNs in locusts, cockroach, cricket, moths, and flies demonstrates that the ventral part of the central brain, including the PS, tritocerebrum, AMMC, and GNG, plays multiple essential roles in triggering insect behaviors. However, the distribution and size of DN somata show no apparent rules concerning their function in behavioral regulation.

As in other species, some long-fiber sensory neurons from peripheral sensory organs and the axons of ascending neurons were stained simultaneously with that of DNs in *H. armigera*. These long-fiber sensory neurons originate from different sensory nerves, including the antennal nerve, frontal ganglion connective nerve, labral nerve, maxillary nerve, and labial nerve. The ascending neurons originate from the thoracic ganglia. The long-fiber sensory neurons and ascending neurons may serve as the fast-speed transmission of sensory information between the brain and the thoracic center. A soma of the long-fiber neuron, here named DNx, which seems to be homologous to the DNx in *D. melanogaster*, was stained near the entrance of the antennal nerve entering into the brain. Such neurons were also found in many other species, for instance, *L. migratoria*, *S. gregaria*, bugs *Rhodnius prolixus* and *Apolygus lucorum*, *D. melanogaster*, and armyworm *Mythimna separata* [52,53,54,55,56].

## 5. Conclusions

The somata of DNs in *H. armigera* stained using backfilling from the neck connective are distributed into different clusters around the surface of the central brain. In general, the DN clusters are conservative across insect species, as was indicated in a comparison of the studied species with, for instance, the locust, cockroach, cricket, flies, and silkmoth. These species are from both hemimetabolous and holometabolous groups. However, the cluster of DNd in *H. armigera* was not found in other studied species, and this cluster was only observed in males. This result suggests that the novel DNd cluster is species- and sex-specific in descending neurons. In addition, the number of the somata of different clusters varies from species, which may indicate that there are some DNs showing species-specific features as well. By comparing the labeling results of both mass and individual filling from other species, we could infer that the ventral part of the central brain, including the PS, tritocerebrum, AMMC, and GNG, plays multiple essential roles in triggering insect behaviors. The DNa, DNd, and DNv clusters might play essential roles in sexually dimorphic behaviors. Therefore, the results of the present study provide a fundamental basis for further studies on the neural mechanism of behavioral responses to the sex pheromone of male *H. amrigera*.

## Figures and Tables

**Figure 1 insects-14-00063-f001:**
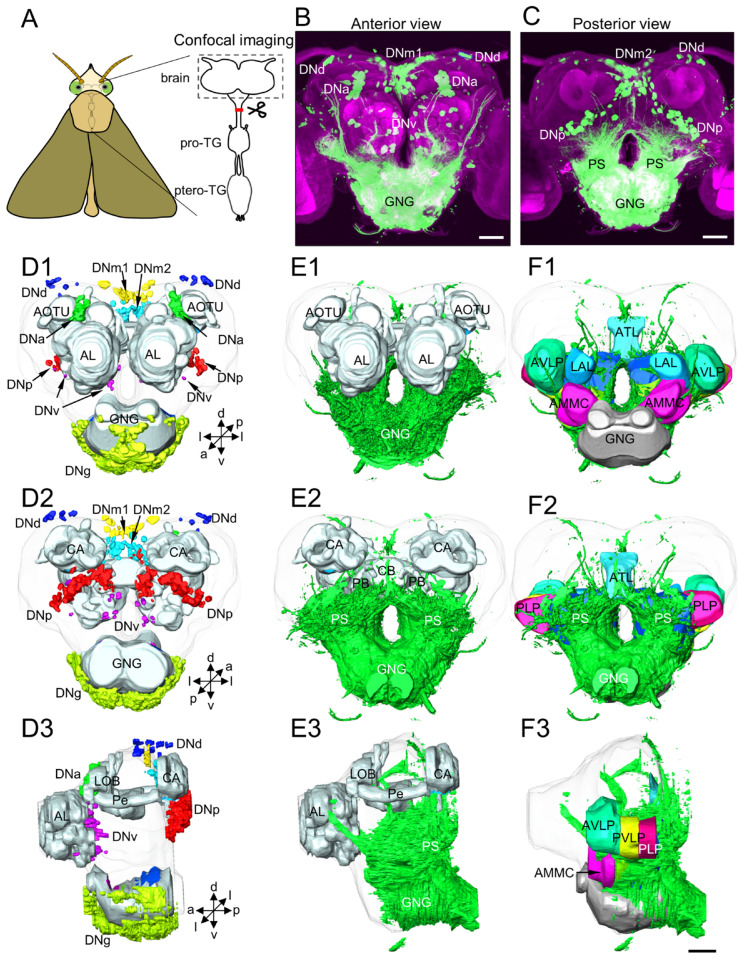
Basic anatomy of descending neurons (DNs) and major innervation areas in the brain of *H. armigera*: (**A**) schematic of backfilling from the neck connective of *H. armigera*; (**B**) anterior view of the brain labeled by backfilling; (**C**) posterior view of the brain labeled by backfilling; (**D1**–**D3**) anterior, posterior, and lateral views of three-dimensional reconstruction showing the distribution of distinct DN somata in the brain; (**E1**–**E3**) anterior, posterior, and lateral views of three-dimensional reconstruction showing DNs innervation in the brain.; and (**F1**–**F3**) anterior, posterior, and lateral views of three-dimensional reconstruction showing different neuropil regions in the brain labeled by backfilling. AL, antennal lobe; AMMC, antennae mechanosensory and motor center; AOTU, anterior optic tubercle; ATL, antler; AVLP, anterior ventrolateral protocerebrum; CA, calyx; CB, central body; GNG, gnathal ganglion; LAL, lateral accessory lobe; LOB, mushroom body lobes; PB, protocerebral bridge; Pe, pedunculus; PLP, posteriorlateral protocerebrum; pro-TG, prothoracic ganglion; PS, posterior slope; ptero-TG, pterothoracic ganglion; PVLP, posterior ventrolateral protocerebrum. a, anterior; d, dorsal; l, lateral; p, posterior; v, ventral. Scale bars: 100 µm.

**Figure 2 insects-14-00063-f002:**
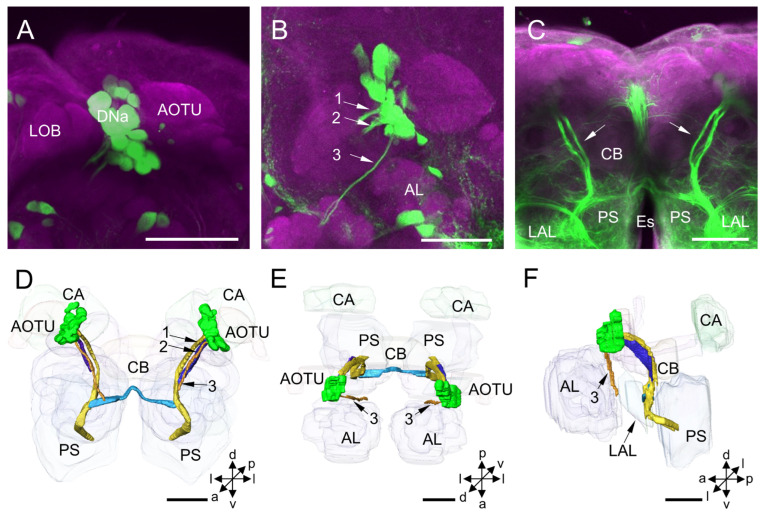
Distribution and projection patterns of DNa neurons in the brain: (**A**–**C**) projection views of confocal stack images showing DNa somata and primary neurites (arrows, No. 1–3); and (**D**–**F**) the anterior, dorsal, and lateral views of the three-dimensional reconstruction showing DNa somata and primary neurites in the brain. AOTU, anterior optic tubercle; AL, antennal lobe; CA, calyx; CB, central body; LAL, lateral accessory lobe; Es, esophagus; LOB, mushroom body lobes; PS, posterior slope. a, anterior; d, dorsal; l, lateral; p, posterior; v, ventral. Scale bars: 100 µm.

**Figure 3 insects-14-00063-f003:**
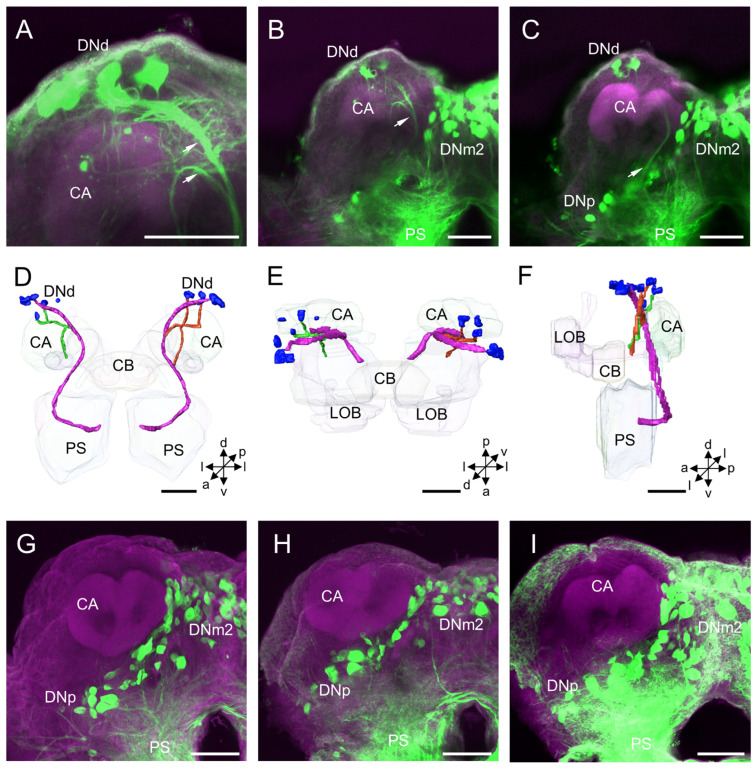
Distribution and projection patterns of DNd neurons in the brain: (**A**–**C**) projection views of confocal stack images showing DNd somata and primary neurites (arrows) in the brain of male *H. armigera*; (**D**–**F**) the anterior, dorsal, and lateral views of the three-dimensional reconstruction showing DNd somata and primary neurites in the brain of male *H. armigera*; and (**G**–**I**) Three preparations of the female brain were successfully labeled by backfilling, showing no stained DNd neurons in the female brain. CA, calyx; CB, central body; LOB, mushroom body lobes; PS, posterior slope. a, anterior; d, dorsal; l, lateral; p, posterior; v, ventral. Scale bars: 100 µm.

**Figure 4 insects-14-00063-f004:**
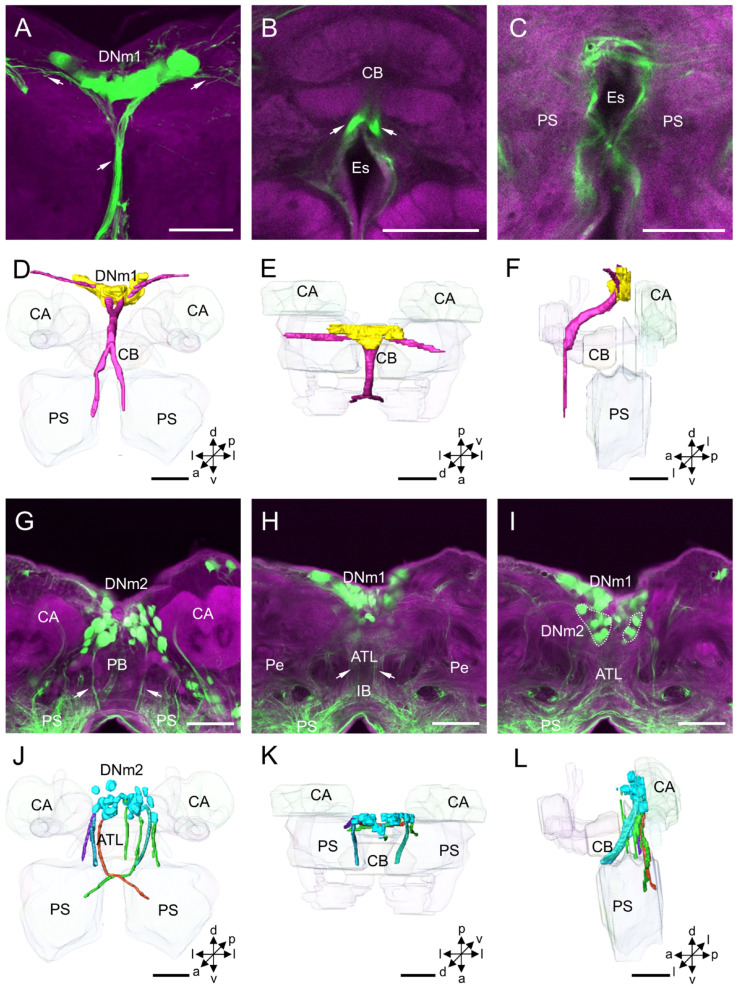
Distribution and projection patterns of DNm neurons in the brain: (**A**–**C**) projection views of confocal stack images showing DNm1 somata and primary neurites (arrows); (**D**–**F**) the anterior, dorsal, and lateral views of the three-dimensional reconstruction showing DNm1 somata and primary neurites in the brain; and (**G**–**I**) projection views of confocal stack images showing DNm2 somata and primary neurites (arrows). (**J**–**L**) the anterior, dorsal, and lateral views of the three-dimensional reconstruction of DNm2 somata and primary neurites in the brain. ATL, antler; CA, calyx; CB, central body; Es, esophagus; IB, inferior bridge; PS, posterior slope. a, anterior; d, dorsal; l, lateral; p, posterior; v, ventral. Scale bars: 100 µm.

**Figure 5 insects-14-00063-f005:**
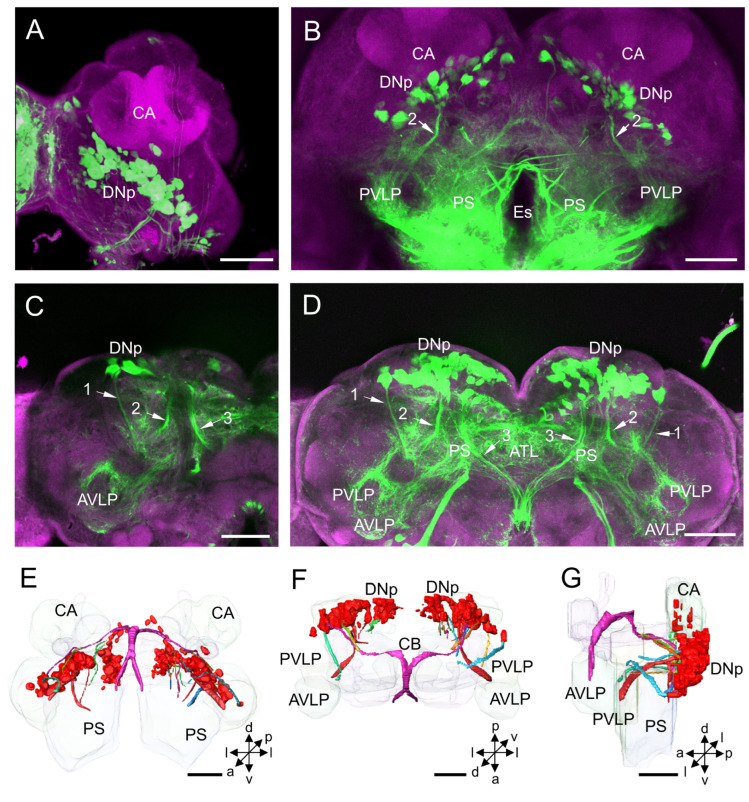
Distribution and projection patterns of DNp neurons in the brain: (**A**,**B**) projection views of confocal stack images of DNp somata and primary neurites (arrows, No. 1–3); (**C**,**D**) projection views of confocal stack images showing DNp somata and primary neurites (arrows); and (**E**–**G**) the anterior, dorsal, and lateral views of the three-dimensional reconstruction showing DNp somata and primary neurites in the brain. AVLP, anterior ventrolateral protocerebrum; CA, calyx; CB, central body; Es, esophagus; PVLP, posterior ventrolateral protocerebrum; PS, posterior slope. a, anterior; d, dorsal; l, lateral; p, posterior; v, ventral. Scale bars: 100 µm.

**Figure 6 insects-14-00063-f006:**
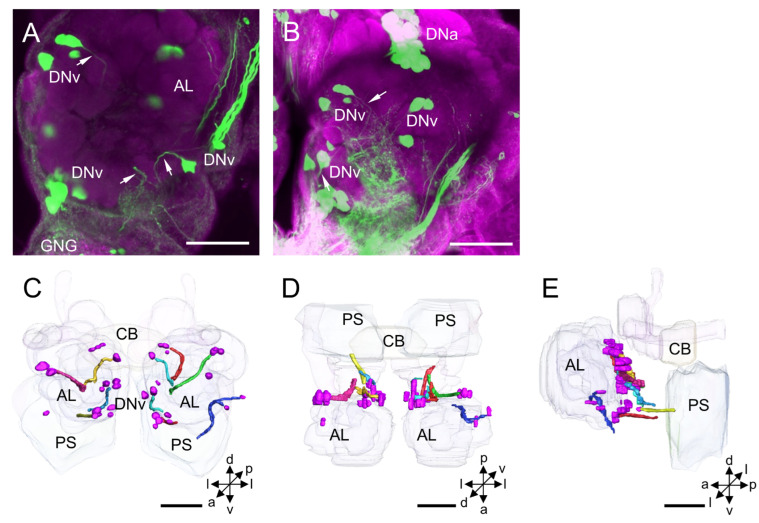
Distribution and projection patterns of DNv neurons in the brain: (**A**,**B**) projection views of confocal stack images showing DNv somata and primary neurites (arrows); and (**C**–**E**) the anterior, dorsal, and lateral views of the three-dimensional reconstruction showing DNv somata and primary neurites in the brain. AL, antennal lobe; CB, central body; GNG, gnathal ganglion; PS, posterior slope. a, anterior; d, dorsal; l, lateral; p, posterior; v, ventral. Scale bars: 100 µm.

**Figure 7 insects-14-00063-f007:**
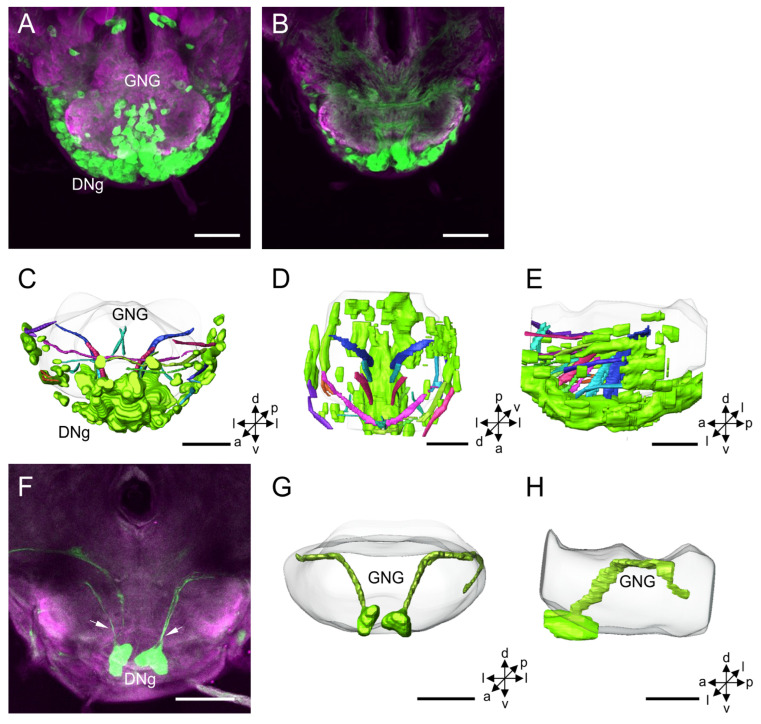
Distribution and projection patterns of DNg neurons in the brain: (**A**,**B**) projection views of confocal stack images showing DNg somata and primary neurites; (**C**–**E**) the anterior, dorsal, and lateral views of the three-dimensional reconstruction showing DNg somata and primary neurites; (**F**) projection views of confocal stack images showing somata in the medial of DNg cluster and their primary neurites (arrows); and (**G**,**H**) the anterior and lateral views of the three-dimensional reconstruction of cells with somata in the medial of DNg cluster and their primary neurites in the brain. GNG, gnathal ganglion. a, anterior; d, dorsal; l, lateral; p, posterior; v, ventral. Scale bars: 100 µm.

**Figure 8 insects-14-00063-f008:**
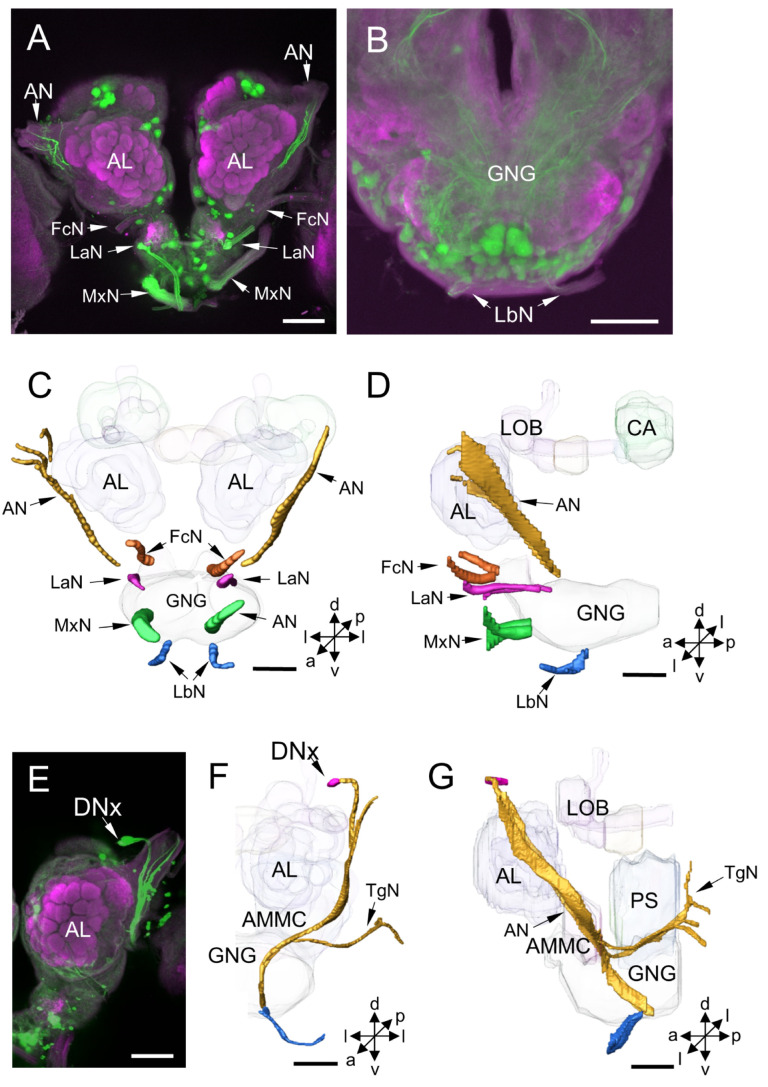
Peripheral nerves of *H. armigera* were labeled by backfilling: (**A**,**B**) projection views of confocal stack images of peripheral nerves by backfilling; (**C**,**D**) the anterior and lateral views of the three-dimensional reconstruction of peripheral nerves; (**E**) projection views of confocal stack images of antennal nerve with two stained somata DNx; and (**F**,**G**) the anterior and lateral views of the three-dimensional reconstruction showing DNx somata and primary neurites in the brain. AL, antennal lobe; AMMC, antennae mechanosensory and motor center; AN, antennal nerve; CA, calyx; FcN, frontal ganglion connective nerve; GNG, gnathal ganglion; LOB, mushroom body lobes; LaN, labral nerve; LbN, labial nerve; MxN, maxillary nerve; PS, posterior slope; TgN, tegumentary nerve. a, anterior; d, dorsal; l, lateral; p, posterior; v, ventral. Scale bars: 100 µm.

**Figure 9 insects-14-00063-f009:**
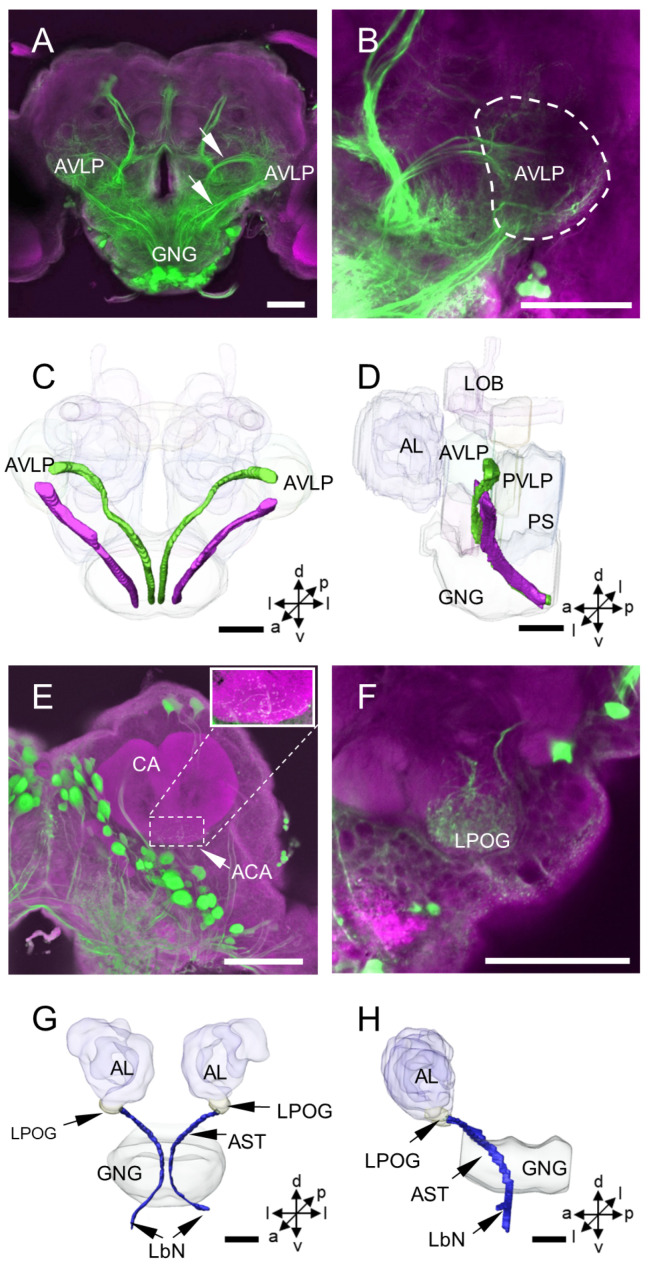
Neurite innervation by DNs and ascending neurons in the brain: (**A**) projection views of confocal stack images of neurite innervation by DNs and ascending neurons labeled in the brain by backfilling, arrows indicate two prominent ascending neuron axon bundles; (**B**) projection views of confocal stack images showing the target areas of two ascending neuron axon bundles; (**C**,**D**) the anterior and lateral views of the three-dimensional reconstruction of two ascending neuron axon bundles in the brain; (**E**) projection views of confocal stack images showing neurites labeled in the ACA by backfilling from the neck connective; (**F**) projection views of confocal stack images showing neurites labeled in the LPOG of the AL by backfilling from the neck connective; and (**G**,**H**) the anterior and lateral views of the three-dimensional reconstruction showing two fibers connecting the LPOG in the antennal lobe of the brain. AL, antennal nerve; ACA, accessory calyx; AST, antenno-subesophageal tract; AVLP, anterior ventrolateral protocerebrum; CA, calyx; GNG, gnathal ganglion; LbN, labial nerve; LOB, mushroom body lobes; LPOG, labial-pit organ glomerulus; PS, posterior slope; PVLP, posterior ventrolateral protocerebrum. a, anterior; d, dorsal; l, lateral; p, posterior; v, ventral. Scale bars: 100 µm.

**Table 1 insects-14-00063-t001:** The number of descending neurons in the brain of *Helicoverpa armigera* and other insect species.

This Study							*Drosophila melanogaster* (Female) [17]			*Periplaneta americana* (Male) ** [16]			*Gryllus bimaculatus* (Female) ** [15]		
Cluster	*H. armigera* (Male)			*H. armigera* (Female)			Cluster Name	Med	Max	Cluster Name	Med	Max	Cluster Name	Med	Max
	**Mean ± SD**	**Med**	**Max**	**Mean ± SD**	**Med**	**Max**									
DNa *	18.60 ± 2.07	19	21	19.00 ± 2.00	19	21	AOTU *	19	38	i5, i5n	23	35	i5, i5n	10	22
DNd *	12.20 ± 2.39	12	15	-	-	-	-	-	-	-	-	-	-	-	-
DNm1	28.40 ± 7.44	28	40	26.33 ± 4.73	28	30	PI	26	48	PI	2	6	PI	5	17
DNm2	31.40 ± 3.58	32	35	33.67 ± 1.53	34	35	-	-	-	i4, c4	18	26	i4, c4	12	17
DNp	196.00 ± 17.09	204	214	188.67 ± 15.14	182	206	SMP	280	325	i1-i3, c1-c3	98	163	i1-i3, c1-c3	111	154
DNv *	14.60 ± 1.82	15	17	13.67 ± 1.53	14	15	PENP	18	80	i7(a,b), c7	11	18	c5, c6, i6, i7(a,b)	8	27
DNg	506.80 ± 43.13	489	582	477.33 ± 28.92	483	503	GNG	462	526	-	-	-	-	-	-

* The number of descending neurons was counted in one hemisphere of *H. armigera* and *D. melanogaster*. In this study, N = 5 for male *H. armigera* and N = 3 for females. ** In the cockroach *P. americana* and cricket *G. bimaculatus* experiments, only one neck connective was labeled. “-” no data.

**Table 2 insects-14-00063-t002:** Diameter and volume of somata of descending neurons in the brain of *Helicoverpa armigera*.

Cluster	Diameter (µm)				Volume (×10^2^ µm^3^)			
	Range		Mean ± SD		Range		Mean ± SD	
	Male	Female	Male	Female	Male	Female	Male	Female
DNa	9.96–24.65	9.33–22.07	16.21 ± 3.65	15.88 ± 3.54	12.50–45.18	11.08–41.71	22.85 ± 8.50	22.04 ± 7.62
DNd	8.24–15.45	-	11.57 ± 1.77	-	7.38–19.73	-	12.16 ± 3.30	-
DNm1	7.13–18.85	8.30–17.87	11.91 ± 2.73	11.27 ± 2.14	6.42–27.65	5.76–18.15	13.35 ± 4.54	10.66 ± 2.85
DNm2	8.09–19.19	9.37–17.33	13.10 ± 2.96	12.65 ± 1.83	5.47–34.18	7.63–50.36	17.07 ± 8.95	15.48 ± 9.17
DNp	8.51–18.11	8.65–16.81	12.46 ± 1.98	11.94 ± 2.35	8.07–25.46	6.61–31.28	15.63 ± 4.90	14.08 ± 5.95
DNv	8.60–20.43	8.41–18.65	13.68 ± 3.33	13.00 ± 2.27	7.34–33.34	8.16–32.57	18.01 ± 6.55	16.98 ± 5.57
DNg	8.44–20.65	9.37–17.07	14.13 ± 3.27	13.14 ± 2.32	6.22–35.10	11.25–31.96	19.86 ± 8.13	17.65 ± 5.59

N = 10 for each cluster in both sexes.

## Data Availability

The original contributions presented in this study are included in the article, further inquiries can be directed to the corresponding authors.

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
