# Peer review of "Distribution and Organization of Descending Neurons in the Brain of Adult Helicoverpa armigera (Insecta)"

_insects, 2023, doi:10.3390/insects14010063_

Round 1

Reviewer 1 Report

The manuscript by Liu et al. is a valuable contribution to a better understanding of descending brain neurons in insects. It is, aside from the fly Drosophila, only the second holometabolous species for which a comprehensive analysis of neurons descending from the brain to the thoracic ganglia has been performed. Being an important pest insect, a better understanding of motor control pathways, as provided here, is highly appreciated. A highlight of this manuscript is clearly the discovery of a sex-specific and perhaps even species-specific population of descending neurons in males that is not present in females. Having said this, I need to emphasize, however, that the manuscript needs overall improvement in language. In my specific comments I have pointed out only a fraction of instances that need to be improved in style and or grammar.

Specific comments

Line 15: “stereotypic”, not stereotypically

Line 17/18: ….candidate neurons that may correlate….

Line 20: …seem to be conserved across….

Line 21/22: …only observed in males….

Line 22: ….DNd cluster consists of species- and….

Line 24 and several other places: the term “midbrain” should be avoided throughout the manuscript as it denotes a particular part of the vertebrate brain. The corresponding term here would be “cerebrum” including proto-, deuto-, and tritocerebrum, or “central brain” which also includes the gnathal ganglion. See Ito et al. 2014, Neuron, vol 81 for terms and nomenclature.

Line 50: ..responding…

Line 69: …by compared with the genetic mark labeling [17]. Poor grammar!

Line 92: …to the plastic pipette with wax.”

Line 92: it should be plural here (scales)

Line 93: what is the “connective nerve”? Your probably mean the neck connectives. Please indicate here that both neck connectives (the right and the left one) were backfilled

Line 105: … with an anti-synapsin antibody…

Line 112: “Followed six times rinses….” Poor style

Line 119: More important than the number of pixels (1024 x 1024) is the pixel size!

Line 144: I do not see the tritocerebrum in Figure 1

Line 145: ..in which there is not much obvious labeling found.” Poor language

Line 148: …showed less or no arborizations were stained…”. There are two verbs in this sentence.

Line 156/157: …showing different neuropil regions in the brain labeled by backfilling.”

Line 165: …are located in the anterior protocerebrum…

Line 208: …ocellar nerve entering the brain…

Line 228: “bifurcate overthere.” Poor language

Line 231: The DNm cluster is located…..

Line 251: The neurites of DNm1 neurons form the medial bundle…. Unclear, do you mean “median bundle”?

Line 279/280: poor language, sentence contains two verbs.

Line 285: The DNv somata should not be described as “DNv cluster” because they do form a single cluster but are widely dispersed at the base of the brain.

Line 303: Likewise, the DNg somata do not form a single cluster, but are widely dispersed along the surface of the GNG.

Line 320: (Figure 7B, F).

Line 327: should it rather be:….observed in the brain, but may be located…?

Line 330: tegumentary nerve

Figure 9E: in my printout of this page I could not see any labeling in the ACA. I suggest enlarging the image of Figure 9E.

Line 353/354: ….backfilling in the brain…. This is misleading! The backfill was from the neck connectives and not in the brain.

Line 408-412: Poor language

Line 420/421: “In addition, neurons homologous to the DNd of H. armigera were not found in the fruit fly, cockroach, and cricket.”

Line 424: Perhaps better: “…DNd neurons are involved in male-specific olfaction.”

Line 438: what is the middle bundle? Do you mean “median bundle”?

Line 490: …have large cell bodies…”

Line 492: “…receiving the stimulation of ocellar illuminations.” Poor style

Line 506: What is the P1 neuron?

Line 509/510: perhaps better: …in controlling backward walking when the fly encouters and impassable barrier

Line 538: perhaps better:”…medial and lateral cell body rind of the GNG.”

Reviewer 2 Report

It is a meticulous analysis of descending neurons revealed from the neck backfilling in Helicoverpa armigera (Ha). 

However, there needs to be a more fundamental understanding of the organization of CNS in insects, so I send the author to read a few papers and incorporate them into their manuscript. 

What is the midbrain in an insect? It would be more appropriate to name it proto, deuto  ... and trito cerebral ganglia. I would send the author to read the book of Bullock and Horridge 1965 about the organization of CNS in insects and begin the introduction as a description of the CNS of Ha. I mean by that, how brain of Ha is organized proto- deuto- trito-. The introduction needs to include how the CNS of Ha is organized: how many fused ganglia are in the tritocerebrum, for example.

Next, about descending neurons in the cockroach, I send you the citation Harrow et al.J Exp Biol., 1980 34-5, where more than one descending neuron is described. That need to be cited.

Other concerns are the Ventral/Dorsal unpaired neurons that must be labeled when you fill in the neck connectives. Check the paper Thompson & Siegler, 1991 J. Comp Neurol 305:659-675 for grasshopper

Round 2

Reviewer 2 Report

The revised manuscript is suitable for publication. I accept the rebuttal in the cover letter.